# A Numerical Study of Geomorphic and Oceanographic Controls on Wave-Driven Runup on Fringing Reefs with Shore-Normal Channels

**Curt D. Storlazzi** [1,*] **, Annouk E. Rey** [2,3,4] **and Ap R. van Dongeren** [3,5]

1   U.S. Geological Survey, Pacific Coastal and Marine Science Center, Santa Cruz, CA 95060, USA
2   Civil Engineering and Geosciences, Delft University of Technology, 2629 HV Delft, The Netherlands; annoukrey@gmail.com
3   Unit of Marine and Coastal Systems, Deltares, 2628 CN Delft, The Netherlands; ap.vandongeren@deltares.nl
4   Boskalis, 3356 LA Papendrecht, The Netherlands
5   Department of Coastal & Urban Risk & Resilience, IHE Delft Institute for Water Education, 2611 AX Delft, The Netherlands
*   Correspondence: cstorlazzi@usgs.gov; Tel.: +1-831-460-7521

**Abstract:** Many populated, tropical coastlines fronted by fringing coral reefs are exposed to wave-driven marine flooding that is exacerbated by sea-level rise. Most fringing coral reefs are not alongshore uniform, but bisected by shore-normal channels; however, little is known about the influence of such channels on alongshore variations on runup and flooding of the adjacent coastline. We conducted a parametric study using the numeric model XBeach that demonstrates that a shore-normal channel results in substantial alongshore variations in waves, wave-driven water levels, and the resulting runup. Depending on the geometry and forcing, runup is greater either on the coastline adjacent to the channel terminus or at locations near the alongshore extent of the channel. The impact of channels on runup increases for higher incident waves, lower incident wave steepness, wider channels, a narrower reef, and shorter channel spacing. Alongshore variation of infragravity waves is predominantly responsible for large-scale variations in runup outside the channel, whereas setup, sea-swell waves, and very-low frequency waves mainly increase runup inside the channel. These results provide insight into which coastal locations adjacent to shore-normal channels are most vulnerable to high runup events, using only widely available data such as reef geometry and offshore wave conditions.

**Keywords:** reef; channel; wave; water level; runup; flooding

## 1. Introduction

Fringing coral reefs serve as a natural coastal protection by dissipating a large portion of incident wave energy before reaching the shoreline. Depending on the reef geometry, reported values of wave attenuation vary from 60% to 99% [1,2] but diminishes for high water levels and large wave conditions [3,4], causing potentially high runup and large flooding during weather events.

The protection by coral reefs is important because the consequences of marine flooding of coastal land adjacent to coral reefs are significant. More than 50% of the total population of Pacific Islands lives within 1.5 km of the shoreline [5] and thus most of these population centers are vulnerable to coastal flooding due to their low elevation [6]. Even on high islands, the presence of steep terrain causes the majority of population, infrastructure (housing, businesses, ports, airports, power plants, sewage treatment plants, etc.), and economic activity to be concentrated on a narrow strip along the coastline susceptible to marine flooding. Locations where the reef is bisected by a shore-normal channel extending from the fore reef to the shoreline (Figure 1) are especially attractive for population centers, as channels provide a natural harbor, easy access to deeper water for vessels, and drinking

water supply through streams. These locations may not be protected by the fringing reefs due to the channel incisions.

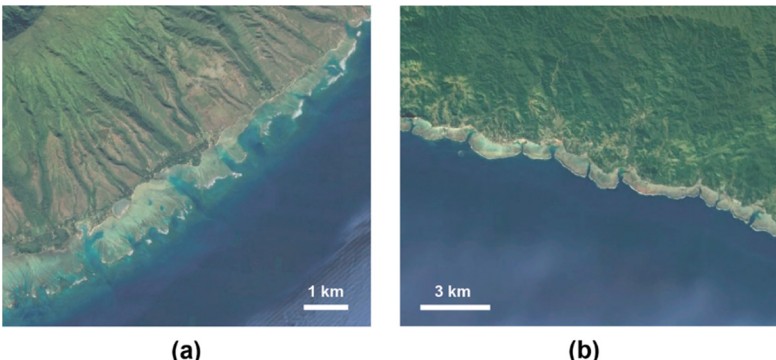

**Figure 1.** Examples of shore-normal channels in fringing coral reef platforms. (**a**) Moloka'i, Hawai'i. (**b**) Viti Levu, Fiji.

Research is ongoing on this subject to understand which hydrodynamic processes lead to these high runup levels, and some valuable insights have been gained [7–9]: the recent increases in oceanic flooding along reef-lined coasts are primarily due to high offshore water levels coinciding with high energy wave events [3,9,10], circumstances which will become more frequent with sea-level rise [11,12]. Increasing water depth over the reef changes the hydrodynamics across the reef in such a way that larger waves reach the shoreline [13], causing high runup levels and flooding of the land behind the shoreline [3,4].

However, most previous research on runup has been based on one-dimensional (1-D) schematization of coral reefs [3,7,10], representing a uniform and straight coastline, whereas most reefs have significant alongshore bathymetric variability. Alongshore variations on coral reef morphology lead to mean currents and circulation that affect nutrient transport [14,15] and hence coral development. Furthermore, mean currents and processes such as wave diffraction and refraction change the directional wave spectrum and can affect alongshore sediment transport [16,17] and wave-driven runup. Previous studies that have investigated alongshore variations in reefs [18–20] usually addressed barrier reef-lagoon systems rather than the more common fringing reefs, and focused on setup differences and flushing times rather than the waves, water levels, and resulting runup at the shoreline. The one study that specifically investigated two-dimensional (2-D) wave processes on fringing reef platforms [21] used a sea-swell ("S", periods < 25 s) wave model that did not include the effects of longer infragravity ("IG", 25 s < periods < 250 s) and low-frequency ("VLF", periods > 250 s) waves, which are important for wave-driven water levels and runup on such reef morphologies [3,4,8].

As of this time, no studies have been performed on the influence of shore-normal reef channels on flooding along fringing reef-lined coasts, specifically during extreme wave conditions when the risk for coastal flooding and the resulting impact to coastal communities is greatest. To address this knowledge gap, we used the physics-based numerical model, XBeach, which has previously been calibrated for fringing coral reefs, to conduct a parametric investigation of how variations in the reef and shore-normal channel morphology and oceanographic forcing influence waves, wave-driven water levels, and the resulting runup on fringing reef coasts.

## 2. Materials and Methods

### 2.1. Reef and Channel Morphologic Parameters

On the basis of Google Earth images, Rey [22] analyzed 70 fringing reef sections that had shore-normal channels and recorded their reef width, channel width, channel spacing, and channel angle. Reef width ($W_r$) was measured next to the channel, perpendicular to the coastline (Figure 2, Appendix A Table A1). Most reefs have a $W_r$ of approximately

300 m, which was the value in the baseline scenario for the numerical modeling. However, the standard deviation in the dataset is large, and thus $W_r$, was varied from 100 to 1000 m between simulations. Channel width ($W_c$) was the average channel width of the channel, perpendicular to the channel axis. Most channels have a $W_c$ of approximately 100 m, which was the value in the baseline scenario; $W_c$ was varied from 50 to 300 m between simulations. The channel spacing ($S_c$) was measured on the two reef sections on both sides of the channel. Most observed $S_c$ are on the order of 1000 m, which was used as the baseline value. Again, the standard deviation in the dataset is large; thus, $S_c$ was varied from 300 to 2000 m between simulations. In most cases, the channel is perpendicular to the coastline, or less than 5 degrees. To keep the baseline scenario simple, the channel was perpendicular to the coastline. Rey [22] also summarized data from 15 previous studies of fringing reefs and recorded their fore reef ($i_{fore}$) and beach slopes ($i_{beach}$). Based on the most common values, in the baseline scenario, both $i_{fore}$ and $i_{beach}$ had values of 1 in 10. Inside the channel, $i_{beach}$ extended below the waterline, and was thus also 1 in 10.

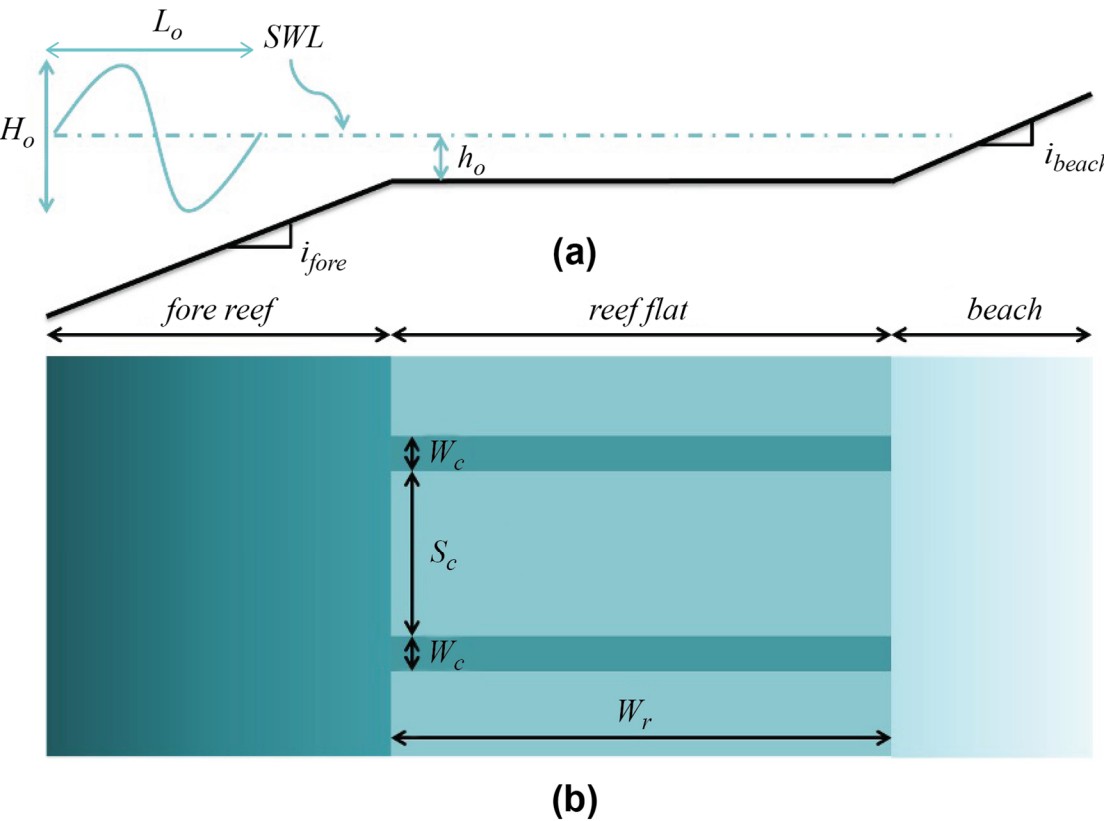

**Figure 2.** Terminology of reef and channel geomorphology and oceanographic forcing. (**a**) Cross-sectional view. (**b**) Map view, with darker colors denoting greater water depth. Geomorphic parameters include reef width, $W_r$, channel width, $W_c$, channel spacing, $S_c$, fore reef slope ($i_{fore}$), and beach slope ($i_{beach}$). Oceanographic parameters include offshore significant wave height, $H_o$, offshore peak wave period, $T_p$, and reef flat water depth, $h_o$, relative to the still water level, *SWL*.

*2.2. Oceanographic Forcing Parameters*

Most atoll and fringing reefs are exposed to both episodic tropical cyclones and large swell events, but the latter are responsible for the majority of documented past high runup events and flooding of reef-lined coasts [4,9]. The primary interest of this study was not the weather extremes that only occur once in a lifetime, but rather episodic events that occur approximately yearly and are a regular nuisance to inhabitants of coastal regions protected by atoll and fringing reefs. Therefore, yearly maxima of 30 years of hindcasted wave conditions developed by Shope et al. [23] were used as the forcing in this study. For the baseline scenario, a significant offshore wave height ($H_o$) of 5 m was applied with a wave

steepness (ratio of $Ho$ to offshore wavelength, $Lo$, thus $Ho/Lo$) of 0.02, corresponding to a peak wave period ($T_p$) of 12.8 s based on the average range (Figure 2, Appendix A Table A1). In the simulations, $Ho$ was varied from 3 to 7 m, corresponding to the yearly maximum wave height ±2 standard deviations, covering 95% of naturally occurring values; $Ho/Lo$ was varied from 0.01 to 0.03, corresponding to the maximum and minimum observed values. For simplicity, in the baseline scenario, the waves approached the coast perpendicularly with no directional spreading. In terms of water levels, most atoll and fringing reef flats have still water depths ($h_o$) of 0.3 to 2.0 m, with an average of 1.0 m [3]. The greatest runup elevations (total water levels) are greatest for deeper $h_o$, but runup is driven largely by setup differences, which are largest when $h_o$ is small [3]. To gain insight into both setup and runup, the baseline scenario had an $h_o$ of 1 m, which was not varied between simulations.

A total of 135 simulations were performed: A combination of 9 different oceanographic forcing scenarios ($Ho$ = 3, 5, 7 m and $Ho/Lo$ = 0.01, 0.02, 0.03), each applied to 15 reef geomorphologies: 4 reference scenarios (simulations of reefs with $W_r$ = 100, 300, 500, 1000 m but without a channel), the baseline geometry, 4 variations in $W_c$ (50, 70, 200, 300 m), 3 variations in $W_r$ (100, 500, 1000 m), and 3 variations in $S_c$ (300, 600, 2000 m).

*2.3. Numerical Modeling*

2.3.1. Model Overview

To understand the influence of alongshore variations in reef geomorphology on waves, water levels, and runup, the physics-based, 2-dimensional (2D) XBeach Non-Hydrostatic (XB-NH) (Deltares, The Netherlands) model was used. XBeach is a process-based, depth-averaged numerical model, which includes a number of hydrodynamic processes such as short and long wave transformation, wave-induced setup, unsteady currents, and overwash [24–26]. It was originally developed to simulate hydrodynamic and morpho-dynamic processes on sandy coasts during storm events. Subsequently, has been applied to a wider range of modeling situations, such as gravel [27] and vegetated coasts [28]. The XBeach model has successfully been applied to coral reef environments in numerous studies [3,7,29–32].

XBeach has two modes, the phase-averaged XBeach Surf Beat (XB-SB) and XB-NH, the latter being phase resolving [26,33]. XB-NH resolves the wave field on the timescale of individual waves, and the model is capable of accurately resolving the non-linear evolution of the wave field, wave–current interaction, and wave breaking in the surf zone [34]. Because the non-hydrostatic mode solves for both long and short waves on the individual wave scale, it is more accurate regarding individual wave breaking and short wave runup than the hydrostatic mode. Here, we used the computationally more expensive XB-NH, as the focus of this study was runup, and it was shown by [3,32,35] that to accurately reproduce wave runup on coral reef-lined coasts, short wave processes cannot be ignored.

2.3.2. Model Settings

In the baseline scenario, the reef has a $W_r$ of 300 m, with a $W_c$ of 100 m, and a $S_c$ of 1000 m. Cyclic boundary conditions were applied on the lateral boundaries, which translate to a 100 m wide channel in the middle of the grid, with an adjacent reef flat of 500 m alongshore on both sides. For simplicity reasons, the channel has a triangular cross section extending to a depth of the base of the reef at 30 m and sloping offshore at the same gradient as $i_{beach}$. At the offshore boundary, the grid was extended to a depth of 30 m followed by 20 horizontal grid cells for a 30 m flat section before the wavemaker. At the shoreward boundary, the model was extended to +12 m with an $i_{fore}$ of 1:10 to account for very high runup levels. A rectangular grid was applied. The alongshore grid resolution varied between 1 m in the channel to 4 m near the boundaries. The cross-shore grid size had a maximum resolution of 0.5 m at the shoreline, corresponding with a 0.05 m vertical resolution for an $i_{beach}$ of 1:10, and a minimum resolution of 1.5 m farther offshore. With an $i_{beach}$ of 1:10, the vertical resolution was 5 cm at the shoreline. A maximum smoothness, defined as the ratio between cell sizes of two adjacent cells, of 1.05 was applied.

The sediment transport and morphological change were disabled. Bottom friction was accounted for by use of the friction coefficient, $c_f$. For the rough fore reef, a $c_f$ of 0.04 was applied, and for the smoother reef flat and beach, a $c_f$ of 0.002 was used, based on the $c_f$ values used in 2-D XBeach reef studies by [12,30].

All of the input wave time series were derived from the same Jonswap spectrum, and were thus identical for all simulations. The offshore water level was kept constant and uniform alongshore. Those time series were applied consecutively at the offshore boundary. Because the hydrodynamic patterns reached a dynamic equilibrium in 1000 s, a conservative spin-up time of 1500 s was applied. $Ru_{2\%}$, defined as the average runup of the highest 2% of waves, requires a large number of waves reaching the coastline during the simulation for accurate computation. It is common practice to calculate the $Ru_{2\%}$ over 1000 waves [3] so the $Ru_{2\%}$ is the average of the 20 highest waves. After 6 h of runtime, the difference relative to the long-term mean $Ru_{2\%}$ value was less than 20%, meaning that most of the variations had damped out, decreasing the influence of occasional high waves on the $Ru_{2\%}$. Therefore, a balance between ideal results and practical runtime was found at a simulated time containing 1500 waves, corresponding with a simulation time of slightly less than 6 h for a $T_p$ of 12.8 s.

Hydrodynamic results were recorded at the offshore boundary, the last point of a water depth of 30 m, midway on the fore reef at 15 m depth, 20 m in front of the reef crest, at the reef crest, 20 m behind the reef crest, midway on the reef flat, and at the beach toe. To generate runup output, numerical runup gauges recorded the instantaneous runup level at 0.5 s intervals. Runup gauges were placed along the entire shoreline, at 2 m along-shore intervals within 120 m of the channel, and at 4 m intervals along the remaining shoreline.

### 2.3.3. Model Validation

Although it would be ideal to calibrate and validate a numerical model with field or laboratory data, to the best of our knowledge, no suitable data are available for this study: no runup or hydrodynamic data are available for coral reefs with channels. However, two factors mitigate the need to calibrate the numerical model on measured data. First, the goal of this conceptual study was not to replicate one case study perfectly, but instead to gain general insights into how the alongshore runup pattern varies in the presence of a shore-normal channel in the reef depending on the local reef geomorphology and oceanographic forcing, and to relate these patterns to dimensionless parameters. Second, as stated before, XB-NH has been applied numerous times for 1-D and 2-D modeling studies [3,7,26,32]. In those studies, the model was calibrated and validated against field and laboratory data, and performed satisfactorily; thus, XB-NH has proven its applicability to coral reef studies and can be used to explore the effects of shore-normal channels in reefs in a conceptual study.

### 3. Results

For all 135 XB-NH model simulations performed, the results were analyzed for reef hydrodynamics and the resulting runup. First, a description of the general trends in the influence of the channel on waves and setup is presented using the reference simulation of a 300 m wide reef flat with a 100 m wide channel being impacted by 7 m at 15 s waves. Next, a description of the general alongshore trends in runup is presented. To better understand what drives runup differences, the contribution of different frequency components to runup are then discussed. Finally, the influence of varying oceanographic forcing and reef and channel geomorphology on runup is assessed by comparison to simulations without a reef channel.

### 3.1. Influence Channels on Waves and Water Levels

Significant total (SS + IG) wave height, $H_{m0}$, is calculated from the variance of the water level. The influence of the channel is assessed by subtracting $H_{m0}$ from $H_{m0}$ in the reference run without a channel. $H_{m0}$ is largest offshore the reef from the channel, in line with the channel (Figure 3a,b). Waves break in front of the reef crest and continue to decrease towards the shoreline. There is a high alongshore gradient of $H_{m0}$ in the channel.

Comparing simulations with and without a channel, the differences are from −3 to +3 m (±42%), indicating the channel has a large influence on wave transformation. There is a relative increase in $H_{m0}$ inside the channel and a decrease in $H_{m0}$ offshore next to the channel. On the reef flat, the cross-shore difference in $H_{m0}$ is small, but at the shoreline there is an alongshore pattern of decreased $H_{m0}$ mid-reef and in the channel, and an increase in $H_{m0}$ next to the channel.

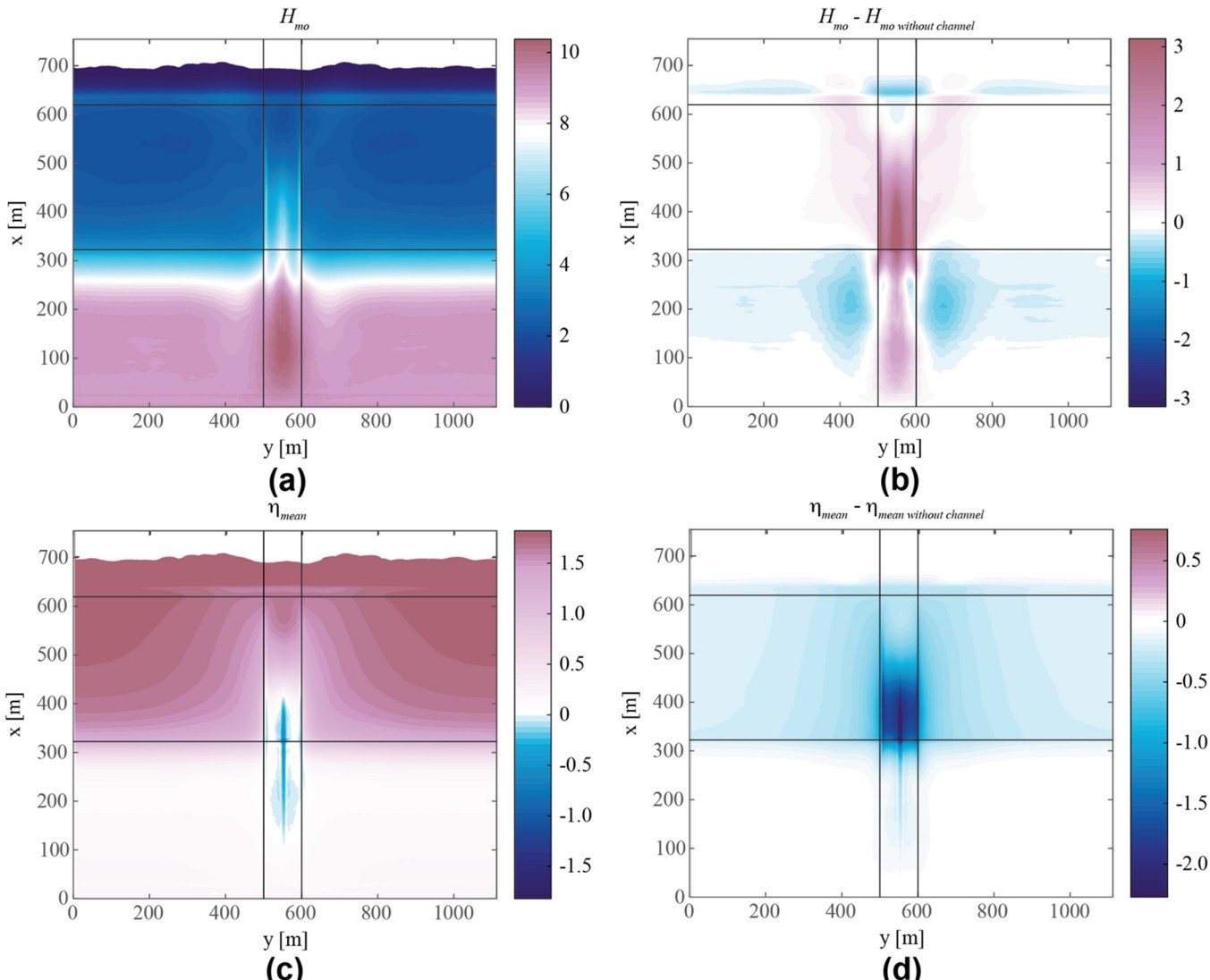

**Figure 3.** Map views of wave height and setup across the reef for large ($Ho$ = 7 m) and long-period ($T_p$ = 15 s) offshore waves. (**a**) Mean wave height, $H_{m0}$. (**b**) Mean wave height, $H_{m0}$, relative to the reference run with the same forcing but without a channel. (**c**) Mean setup, $\eta_{mean}$. (**d**) $\eta_{mean}$, relative to the to the reference run with the same forcing but without a channel. The red zones indicate higher values in the case with a channel as compared to the reference case without a channel, and the blue zones indicate smaller values in the case with a channel. In these plots, offshore is on the bottom at $x$ = 0, the reef crest at $x$ = 320 m, and the still-water shoreline at $x$ = 620 m; the channel extends from $y$ = 500–600 m.

The mean setup ($\eta_{mean}$) is calculated as the mean water level over the entire simulation, and the influence of the channel is assessed by locally subtracting $\eta_{mean}$ from $\eta_{mean}$ in the reference run without a channel. Wave forcing causes $\eta_{mean}$ on the reef flat up to 1.5 m (Figure 3c). In the channel, in the vicinity of the reef crest, $\eta_{mean}$ is lower than the SWL, whereas near the shoreline, the setup is positive, decreasing alongshore from mid-reef

towards the channel. The channel decreases $\eta_{mean}$ up to 2 m (Figure 3d). The difference is greatest in the channel near the reef crest and decreases with increasing distance from the channel. $\eta_{mean}$ increases shoreward from the reef crest towards the beach toe, and alongshore from the channel towards the mid-reef. There is a steep gradient in $\eta_{mean}$ on the channel edge.

Thus, $H_{m0}$ was found to decrease towards the shoreline, whereas $\eta_{mean}$ increases towards the shoreline. Variations of $H_{m0}$ and $\eta_{mean}$ are most notable within 100 m from the channel, whereas the runup pattern displays strong variation for twice that distance. Both $H_{m0}$ and $\eta_{mean}$ have sharp gradients around the channel edge.

### 3.2. Alongshore Patterns in Runup

To assess the influence of the shore-normal reef channel on $Ru_{2\%}$, the scenarios with a channel were compared to the reference runs without a reef channel (Figure 4). There is alongshore variability in $Ru_{2\%}$, with a range of approximately 20% of $Ho$. The lack of perfect symmetry across the channel is due to the irregular SS and IG waves interacting with the reef morphology. The highest $Ru_{2\%}$ (peak in $Ru_{2\%}$) is 5% higher in the simulation with a channel than without, and the lowest $Ru_{2\%}$ (trough in $Ru_{2\%}$) is 15% lower with the channel without. The maxima (peaks) in $Ru_{2\%}$ are approximately 120 m away from the channel edges, the minima (troughs) in $Ru_{2\%}$ are found within approximately 40 m of the channel edges, and there is a local $Ru_{2\%}$ maximum in the middle of the channel. The $Ru_{2\%}$ in the middle of the reef (far from the channel) is 5% lower in the simulation with a channel than in the reference run.

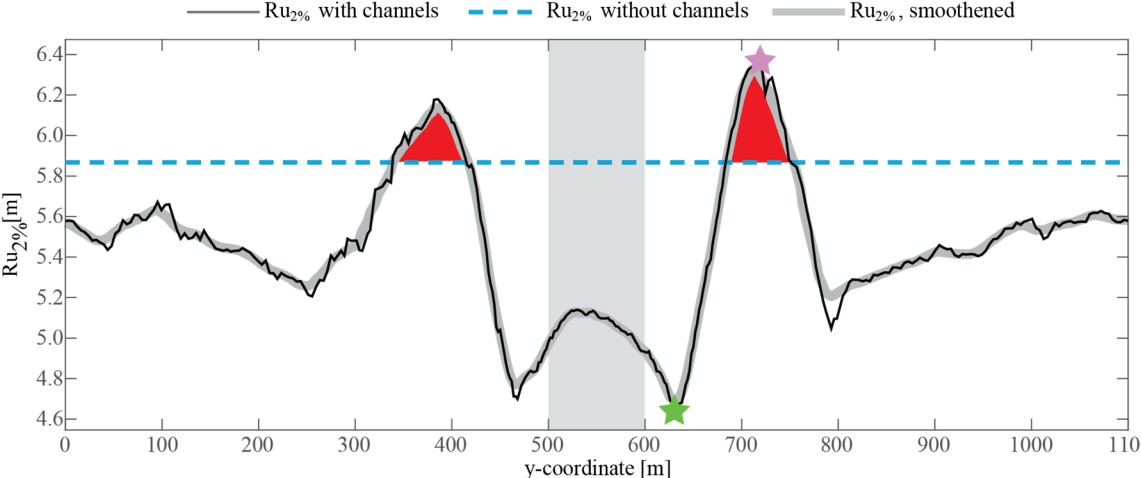

**Figure 4.** Alongshore patterns in runup, $Ru_{2\%}$, for the reference case of a reef flat ($W_r$ = 300 m) without a channel (dashed line) and with a channel (solid line) under identical large ($Ho$ = 7 m) and long-period ($T_p$ = 15 s) offshore waves. High runup zones where $Ru_{2\%}$ exceeds the reference $Ru_{2\%}$ are indicated by the red regions. The peak in $Ru_{2\%}$ is indicated by a purple star; the lowest $Ru_{2\%}$ is indicated by the green star. The gray region denotes the location of the channel ($W_c$ = 100 m).

### 3.3. Frequency Components of Runup

To better understand which wave processes are responsible for the differences in $Ru_{2\%}$ alongshore, the influence of different frequency components on $Ru_{2\%}$ was assessed. For each runup gauge, the vertical runup signal was split into setup (mean over entire run, $Ru_s$), SS ($Ru_{Hmo,SS}$), IG ($Ru_{Hmo,IG}$), and VLF ($Ru_{Hmo,VLF}$) water level time series. From these time series, the runup height of each frequency component was calculated using the variance in the water level time series, which was normalized against $Ho$ (Figure 5).

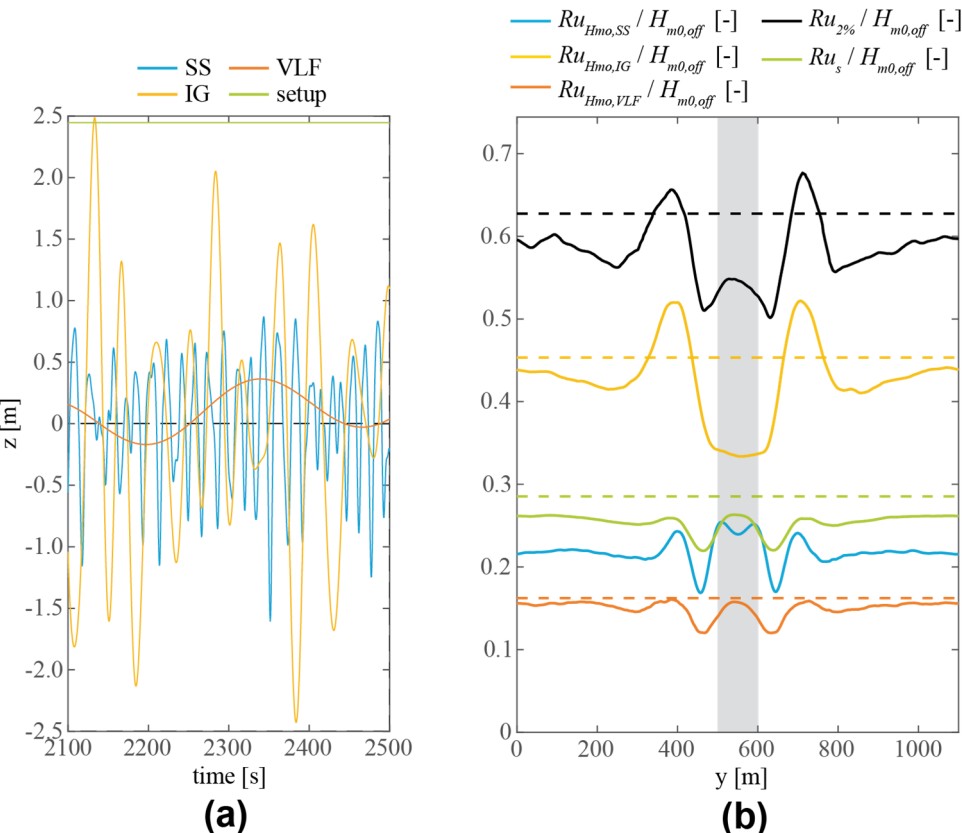

**Figure 5.** Frequency components of waves and runup on the reef flat. (**a**) Time series of the SS (blue), IG (orange), VLF (red), and setup (green) components of water level, taken in the middle of reef by the shoreline. (**b**) Alongshore varying runup ($Ru_{2\%}$, black) split into setup ($Ru_s$, green), SS ($Ru_{Hmo,SS}$, blue), IG ($Ru_{Hmo,IG}$, orange), and VLF ($Ru_{Hmo,VLF}$, red) wave components, presented as the fraction of the incoming waves by normalizing with wave height at the offshore boundary (*Ho*). The dashed lines in corresponding colors show the frequency components in the reference run without a channel. The gray region denotes the location of the channel.

$Ru_{Hmo,IG}$ is dominant, with $Ru_{Hmo,IG}$ up to 50% of the offshore *Ho*, which is approximately double the contribution of the other components. The $Ru_{Hmo,IG}$ peaks align with the $Ru_{2\%}$ peaks, but the local $Ru_{2\%}$ increase in the channel is not present in the $Ru_{Hmo,IG}$. The vertical range is similar to the $Ru_{2\%}$ range. In this simulation, $Ru_s$ is the second largest contributor to $Ru_{2\%}$. The channel decreases $Ru_s$ over the entire alongshore domain, as it is lower than in the reference run. The $Ru_s$ peaks next to the channel align with $Ru_{2\%}$ peaks, and also have a peak inside the channel that is present in the $Ru_{2\%}$ pattern, but not in the $Ru_{Hmo,IG}$. $Ru_{Hmo,VLF}$ is approximately half the $Ru_s$, with a similar alongshore pattern. $Ru_{Hmo,SS}$ only varies from the reference scenario in the vicinity of the channel, with peaks just inside the channel.

### 3.4. Influence of Geomorphology and Oceanographic Forcing on Runup

To assess the influence of the varying oceanographic forcing and reef and channel geomorphology on $Ru_{2\%}$, the maximum $Ru_{2\%}$ of every simulation was compared to the three geomorphic ($W_c$, $W_r$, and $S_c$) and two forcing (*Ho* and *Ho/Lo*) parameters (Figure 6). The range of $Ru_{2\%}$, defined as the difference between the highest $Ru_{2\%}$ peak and the minimum $Ru_{2\%}$ peak, varies for different forcings on the same geometry; its range is approximately 20% of *Ho*. $Ru_{2\%}$ maxima are higher for scenarios with higher *Ho* and low *Ho/Lo*, short $S_c$, narrower $W_r$, and wider $W_c$. Both *Ho* and *Ho/Lo* have a large influence on $Ru_{2\%}$. Of the geomorphic parameters, $W_c$ has the strongest influence on $Ru_{2\%}$. Whereas $Ru_{2\%}$ trends relatively linearly with the forcing (*Ho* and *Ho/Lo*) parameters, there are

relatively non-linear response in $Ru_{2\%}$ with changes in the geomorphic ($W_c$, $W_r$, and $S_c$) parameters, many of which have local minima.

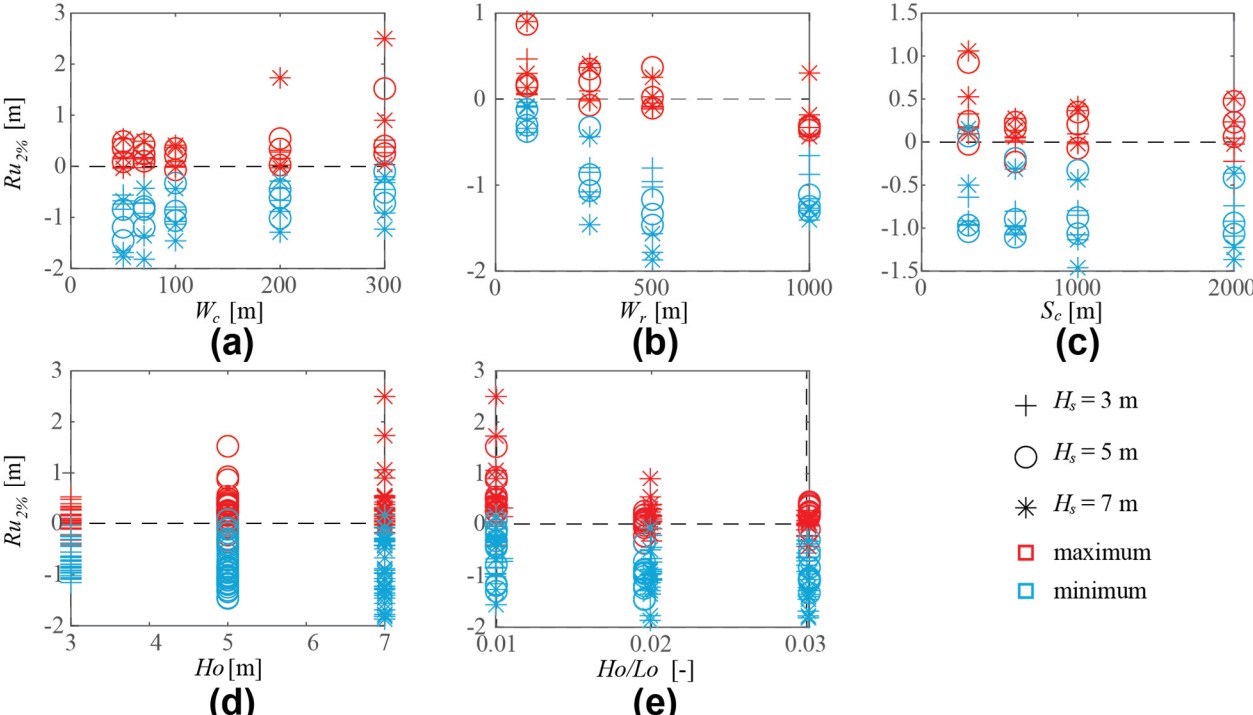

**Figure 6.** Runup, $Ru_{2\%}$, for different reef and channel geomorphic parameters or oceanographic forcing parameters. (**a**) Channel width, $W_c$. (**b**) Reef width, $W_r$. (**c**) Channel spacing, $S_c$. (**d**) Significant wave height, $Ho$. (**e**) Wave steepness, $Ho/Lo$. The marker shapes indicate the $Ho$ and colors the minimum or maximum values for a simulation.

## 4. Discussion

A shore-normal channel in a fringing reef strongly influences the overall $H_{m0}$ and $\eta_{mean}$ patterns on the reef flat and in the channel. Alongshore gradients of $H_{m0}$ and $\eta_{mean}$ are large in the vicinity of the channel. $\eta_{mean}$ increases towards the shoreline and towards mid reef. Differences in $H_{m0}$ are most notable on the fore reef, in the channel, and at the shoreline. $H_{m0}$ and $\eta_{mean}$ just offshore the beach toe have increasing correspondence with the $Ru_{2\%}$ pattern, but still there are significant differences. There is a decrease in $Ru_{2\%}$ in the vicinity of the channel, with a small increase inside the channel, a peak next to the channel, and a gradual approach of the reference $Ru_{2\%}$ towards mid-reef. Sometimes, but not always, a local $Ru_{2\%}$ peak occurs inside the channel.

Variations in both reef geomorphology and oceanographic forcing change one or more of the following characteristics. For small $Ho$ and $Ho/Lo$, the $Ru_{2\%}$ pattern does not show a clear peak, just a decrease in $Ru_{2\%}$ towards the channel. High $Ho/Lo$ and wide $W_c$ both result in a dual $Ru_{2\%}$ peak system with a second $Ru_{2\%}$ peak inside the channel. Narrow $W_r$ make the $Ru_{2\%}$ pattern more alternating and irregular, whereas wide $W_r$ decrease $Ru_{2\%}$ compared to the reference $Ru_{2\%}$ without a channel. Short $S_c$ move the $Ru_{2\%}$ peak towards mid-reef, whereas for long $S_c$, the longshore $Ru_{2\%}$ variation stays restricted to the vicinity of the channel. The largest $Ru_{2\%}$ peaks occur in the channel, for wide $W_c$ and high $Ho$ and $Ho/Lo$. $Ru_{2\%}$ peaks are largest when $W_c$ takes up a large fraction of the total shoreline and when the incident $T_p$ is closer to the resonant period of the reef.

A schematic that illustrates the general $Ru_{2\%}$ pattern, and the impact of each geomorphic or oceanographic parameter on this pattern, is presented in (Figure 7). This schematic illustrates the following conclusions regarding the alongshore variations in the $Ru_{2\%}$ patterns:

1.  For small $Ho$ and $Ho/Lo$, the $Ru_{2\%}$ pattern does not show a clear peak, just a decrease in $Ru_{2\%}$ towards the channel.
2.  Low $Ho/Lo$ and wide $W_c$ both result in a dual $Ru_{2\%}$ peak system, with a second $Ru_{2\%}$ peak inside the channel.
3.  Narrow $W_r$ make the $Ru_{2\%}$ pattern more alternating and irregular, whereas wide $W_r$ decrease $Ru_{2\%}$ compared to the reference $Ru_{2\%}$ without a channel.
4.  Short $S_c$ result in a $Ru_{2\%}$ peak farther mid-reef, whereas for long $S_c$, the alongshore $Ru_{2\%}$ variation is restricted to the vicinity of the channel.
5.  The largest $Ru_{2\%}$ peaks occur in the channel, for wide $W_c$ and high and high $T_p$ waves. $Ru_{2\%}$ peaks are greatest as the channel takes up a large fraction of the total shoreline when the $T_p$ is closer to the resonant period of the reef.

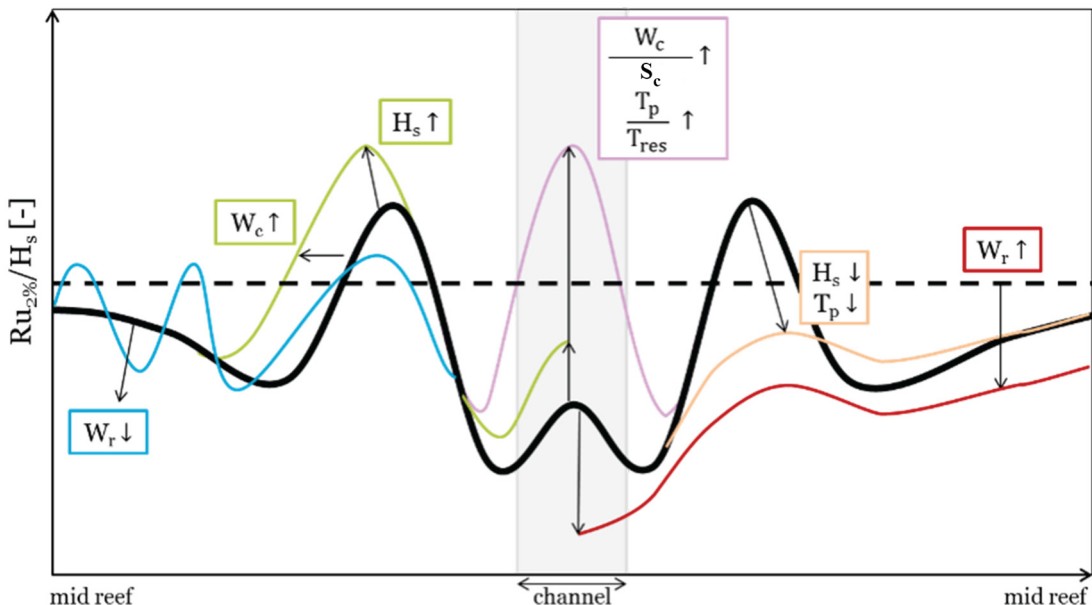

**Figure 7.** Overview of runup pattern for variations in reef and channel geomorphology and oceanographic forcing. The solid black line shows the general alongshore runup pattern for a reef with a channel, and the dashed line represents the reference line for a case without a channel. The colored lines illustrate the influence of variations in forcing and geomorphology on changing patterns in runup relative to the base case (solid black line). Arrows denote the trends in change relative to the base scenario for the given parameter or ratio between the parameters. Portions of solid lines above the dashed line indicate alongshore areas, relative to the channel, where there is greater runup than in the case without a channel, and those below the dashed line indicate those alongshore areas with less runup.

In terms of the relative contribution of frequency components to $Ru_{2\%}$ for varying geomorphologies and oceanographic forcing, IG waves are the dominant contributor. The shape is similar to the $Ru_{2\%}$ pattern, except for the local peak inside the channel, and the contribution is larger than that of the SS, VLF, and setup components. This is in line with findings from 1-D $Ru_{2\%}$ studies on reefs [3,4,7]. For waves with low $Ho$ and high $Ho/Lo$, SS has the second-largest contribution to $Ru_{2\%}$, whereas, for waves with high $Ho$ and low $Ho/Lo$, setup becomes increasingly important. This is in line with expectations, as setup is smaller for increasing relative reef submergence and $Ho/Lo$ [8,18,36]. For simulations where the $Ru_{2\%}$ is largely below the reference $Ru_{2\%}$, setup and VLF are also lower than their reference line, whereas alongshore mean SS and IG are near their reference level. This indicates VLF and setup are responsible for the global decrease in $Ru_{2\%}$ alongshore. Variations in VLF and setup are mostly gradual and extend toward the mid-reef, whereas SS gradients are largest in the vicinity of the channel and do not vary much towards mid-reef; IG are between SS and VLF. This may indicate that the wavelength of a frequency

component is linked to the region of influence from the channel of the given component. Further numerical and field studies may confirm this.

## 5. Conclusions

The results of this numerical model study illustrate the importance of 2-D effects for $Ru_{2\%}$. Most previous modeling studies of coral reefs have been based on 1-D schematizations, and it is expected that future studies will also be, due to the huge difference between computational demand, and hence runtime, of 1-D versus 2-D models. Whether it is justified to use a 1-D transect rather than a 2-D model depends on the distance from the channel.

The current results show that, even far from the channel, even at mid-reef, the channel still has influence on $Ru_{2\%}$ levels, due to the decrease in setup. In the present study, the channel increases the $Ru_{2\%}$ with maximum 5% of $Ho$ away from a channel. In this case, a 1-D schematization of a transect is expected to provide a reasonable estimate of $Ru_{2\%}$ under relatively high $Ho$ and long $T_p$ conditions.

Near a channel (within two offshore wave lengths of the channel), the channel increases $Ru_{2\%}$ locally, and a 1-D schematization may underestimate extreme $Ru_{2\%}$ levels and flood risk. Furthermore, the strong gradients of $H_{m0}$ and $h_0$ (and thus flow velocity, not shown) near the channel indicate the importance of 2-D processes in the region near the channel, which are not accounted for in 1-D models. If one is interested in extreme $Ru_{2\%}$ levels in this region, a 2-D model is recommended. If that is not possible due to, for instance, the high computational demand, and the only option is to use a 1-D transect, a conservative correction factor should be applied to account for the expected increase in $Ru_{2\%}$ due to the presence of the channel. Based on the range of simulations in the present study, $Ru_{2\%}$ on the reef flat near the channel can be up to 20% of $Ho$ higher than without a channel; however, as scatter is large, more 2-D simulations are recommended before determining such a correction factor.

Inside a channel, $Ru_{2\%}$ is lower than without a channel for narrow channels and moderate to high waves, but $Ru_{2\%}$ can be up to 40% of $Ho$ higher than without a channel for wide channels and high and long offshore waves. In those cases, $Ru_{2\%}$ is hypothesized to approach the $Ru_{2\%}$ of a case where there is no protective reef at all. Since in the present study no simulations were performed of beaches without a protective reef, it is unknown whether simulations using 1-D transects would overestimate or underestimate $Ru_{2\%}$ inside the channel. If one is interested in the $Ru_{2\%}$ inside a channel, a 2-D model is therefore recommended.

For a given fringing reef shoreline with a shore-normal channel, predictions can be made regarding the parts of the shoreline that are most likely to receive high $Ru_{2\%}$ levels for different oceanographic forcings. This knowledge can be used for a various range of coastal zone management policy decisions, such as siting and designing new infrastructure or developing evacuation plans and early warning systems [6].

**Author Contributions:** Conceptualization, methodology, software, validation, formal analysis, investigation, data curation, writing—original draft preparation, writing—review and editing, and visualization, C.D.S., A.E.R. and A.R.v.D.; supervision, project administration, and funding acquisition, C.D.S. and A.R.v.D. All authors have read and agreed to the published version of the manuscript.

**Funding:** This research was funded by the U.S. Department of Interior, U.S. Geological Survey through the Coastal and Marine Hazards and Resources Program, the U.S. Department of Defense, Strategic Environmental Research and Development Program's Project RC-2334, and Deltares Strategic Research in the Natural Hazards Program.

**Institutional Review Board Statement:** Not applicable.

**Informed Consent Statement:** Not applicable.

**Data Availability Statement:** Data needed to evaluate the conclusions in the paper are available from [37] at https://doi.org/10.5066/P9A0HFKV (accessed on 14 May 2022).

**Acknowledgments:** We would like to thank Mark Buckley (USGS) for his important insight and useful comments during the preparation of this article. Any use of trade, firm, or product names is for descriptive purposes only and does not imply endorsement by the US Government.

**Conflicts of Interest:** The authors declare no conflict of interest.

## Appendix A

**Table A1.** Reef and channel geomorphic parameters and oceanographic forcing parameters used in this study.

| Parameter [Units] | Definition |
| --- | --- |
| $W_r$ [m] | Reef flat width, distance between the reef crest and the beach toe |
| $W_c$ [m] | Channel width |
| $S_c$ [m] | Channel spacing, or the length of a reef section between two channels |
| $i_{fore}$ [-] | Slope of the fore reef |
| $i_{beach}$ [-] | Slope of the beach |
| $Ho$ [m] | Significant offshore wave height |
| $H_{m0}$ [m] | Mean wave height |
| $H_{m0,SS}$ [m] | Mean sea-swell ('SS') band wave height |
| $H_{m0,IG}$ [m] | Mean infragravity ('IG') band wave height |
| $H_{m0,VLF}$ [m] | Mean very-low frequency ('VLF') band wave height |
| $T_p$ [s] | Peak wave period |
| $Lo$ [m] | Offshore wavelength, proportional to $T_p{}^2$ |
| $Ho/Lo$ [-] | Wave steepness, the ratio between wave height and wavelength |
| $SWL$ [m] | Still water level, $SWL = 0$ by definition |
| $h_0$ [m] | Water depth on the reef flat |
| $\eta_{mean}$ | Mean setup, the mean water level over the entire simulation |
| $Ru_{2\%}$ [m] | The average runup of the highest 2% of waves |
| $Ru_s$ [m] | The contribution of setup to $Ru_{2\%}$ |
| $Ru_{Hmo,SS}$ [m] | The contribution of sea-swell ('SS') band wave height to $Ru_{2\%}$ |
| $Ru_{Hmo,IG}$ [m] | The contribution of infragravity ('IG') band wave height to $Ru_{2\%}$ |
| $Ru_{Hmo,VLF}$ [m] | The contribution of very-low frequency ('VLF') to $Ru_{2\%}$ |

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
