# Peer review of "A Numerical Study of Geomorphic and Oceanographic Controls on Wave-Driven Runup on Fringing Reefs with Shore-Normal Channels"

_jmse, doi:10.3390/jmse10060828_

Round 1

Reviewer 1 Report

This is an interesting paper studying the impact of coral reefs on wave runups. The authors used the XBeach model and performed a series of simulations. In my opinion, the Figure 9 (or Figure 7) of this paper summarizes all the simulation results and has the potential to be a benchmark of the coastal community. Although the simulation results are not validated against experimental data, they are still worth being published. Also, this paper is very well written and the messages are very well delivered. I only have several minor concerns before its publication:

  1. How are the coral reefs parameterized in the XBeach model? Are they treated as non-slip walls?
  2. Line 111, is i_f = i_fore and i_b = i_beach?
  3. The authors should increase the font size for Figure 3. Why are the authors using a different orientation compared with Figure 2?
  4. Why are the patterns shown in Figure 4 not symmetrical? Any comments?
  5. Figure 7 and 8 are missing before Figure 9.

Reviewer 2 Report

The manuscript explains a model try to understand fringing reefs impact on runup. The content is clearly explained. The manuscript is very well written, explains all the questions that a reader can have. A couple sentences to explain what aspects to consider for future studies on the conclusion will be good. I would like to thank authors for this very well written reseach. 

Reviewer 3 Report

Paper title: A numerical study of geomorphic and oceanographic controls on wave-driven runup on fringing reefs with shore-normal channels

Authors: Curt D Storlazzi, Annouk E Rey, Ap R Van Dongeren

Journal: Journal of Marine Science and Engineering, MDPI

General comment

The paper reports on a numerical investigation made with the aid of XBeach model to analyse the effects of a shore-normal channel on a fringing reef coastline. Wave runup, setup and wave height are calculated and compared to the case of channel absence.

The paper lacks a proper calibration and validation but as reported by the same authors, no runup or hydrodynamic data are available for coral reefs with channels, so that this weak point cannot be overcome.

The paper is well organized in the complex and it can be considered for publication once few minor points indicated below have been fixed.

Minor points

Figure 2 and related text: a reference system should be defined and reported in the figure.

Figure 3: please add the ticks to the colorbars of insets b) and d).

Sentence in lines 284-286 is not clear.

Symbols are rather confused, e.g. wave height is indicated sometimes with H0 and other with Hs, as well as wave steepness, which sometimes is referred to as sw and others as H0/L0.

Author Response

The paper reports on a numerical investigation made with the aid of XBeach model to analyse the effects of a shore-normal channel on a fringing reef coastline. Wave runup, setup and wave height are calculated and compared to the case of channel absence.
The paper lacks a proper calibration and validation but as reported by the same authors, no runup or hydrodynamic data are available for coral reefs with channels, so that this weak point cannot be overcome.

REPONSE: Thank you for understanding this. As discussed in the Methods section, the model has been validated for coral reefs elsewhere (see literature cited), and as this is a parametric study, there is no field site and thus ability to have field data for calibration and validation.

The paper is well organized in the complex and it can be considered for publication once few minor points indicated below have been fixed.

Minor points:
Figure 2 and related text: a reference system should be defined and reported in the figure.
REPONSE: It is unclear to a “reference system”, as it is a schematic diagram with the locations of the beach and reef to denote orientation. We did add, “Map view, with darker colors denoting greater water depth” in the figure caption for additional explanation.

Figure 3: please add the ticks to the colorbars of insets b) and d).
REPONSE: There are ticks and values in the colorbar in the figure file (‘Fig_3_Hs_ho_mean_diff.png’) that somehow did not come through in the journal’s version of the manuscript. We re-embedded the correct image into the revised version of the manuscript.

Sentence in lines 284-286 is not clear.

REPONSE: The text is currently, “The peaks in Ru2% are approximately 120 m away from the channel edge, the minima (troughs) in Ru2% are found within approximately 40 m of the channel edges, and there is a local Ru2% maximum in the middle of the channel.”. The text was revised to, “The maxima (peaks) in Ru2% are approximately 120 m away from the channel edges, the minima (troughs) in Ru2% are found within approximately 40 m of the channel edges, and there is a local Ru2% maximum in the middle of the channel.”

Symbols are rather confused, e.g. wave height is indicated sometimes with H0 and other with Hs, as well as wave steepness, which sometimes is referred to as sw and others as H0/L0.

REPONSE: We has switched offshore significant wave height from “Hs” to “Ho” and missed one instance – thank you for catching that. We replaced the “Hs” with “Ho”. The x-axis label of Figure6e in the figure file (‘Fig_6_scatterplot_Ru2_vs_all_parameters.png’) is ‘Ho/Lo’, not ‘sw’, that somehow did not come through in the journal’s version of the manuscript. We re-embedded the correct image into the revised version of the manuscript.